# The Effect of Environmental Factors on the Diversity of Crane Flies (Tipulidae) in Mountainous and Non-Mountainous Regions of the Qinghai-Tibet Plateau and Surrounding Areas

**DOI:** 10.3390/insects13111054

**Published:** 2022-11-15

**Authors:** Qicheng Yang, Wei Chen, Lishan Qian, Ding Yang, Xiaoyan Liu, Manqun Wang

**Affiliations:** 1Hubei Insect Resources Utilization and Sustainable Pest Management Key Laboratory, College of Plant Science & Technology, Huazhong Agriculture University, Wuhan 430070, China; 2CAS Key Laboratory for Plant Diversity and Biogeography of East Asia, Kunming Institute of Botany, Chinese Academy of Sciences, Kunming 650201, China; 3Department of Entomology, College of Plant Protection, China Agricultural University, Beijing 100193, China

**Keywords:** Asia, climatic factors, endemism, prediction model, species richness, topographic heterogeneity

## Abstract

**Simple Summary:**

Understanding the macro pattern and underlying mechanism of species diversity is critical for research on biodiversity and biological conservation. However, there have been few reports about the different effects of the same environmental factor on biodiversity between mountainous and non-mountainous regions. This study revealed the diversity patterns of Tipulidae in the Qinghai-Tibet Plateau and its surrounding areas, investigated the influence of environmental factors on its diversity in mountainous and non-mountainous regions, and deduced the richness model of Tipulidae in mountainous regions. Our results revealed three highly endemic regions and provided a species richness model of a group of decomposer insects. The warmest quarter precipitation and topographic heterogeneity have main effects on the diversity of Tipulidae in mountainous regions. These findings provide a reference for the diversity model of decomposers to aid in the protection of biodiversity in Asian mountains.

**Abstract:**

Tipulidae, one of the most diverse families of Diptera, is widely distributed in the world. The adults have weak flight ability, making it an ideal model for studying the formation of insect diversity. This study aims to explore the species diversity and endemism of Tipulidae in the Qinghai-Tibet Plateau and the surrounding areas, as well as analyze the relationships between the diversity pattern and 25 environmental factors in mountainous and non-mountainous regions. To this end, we collected 2589 datasets for the distribution of 1219 Tipulidae species, and found three areas with high diversities of Tipulidae around the QTP, including the Sikkim-Yadong area, Kamen River Basin, and Gongga Mountain. Further R, generalized additive model (GAM), and stepwise multiple regression analysis indicated that the richness and endemism of Tipulidae is mainly influenced by the warmest quarter precipitation and topographic heterogeneity in mountainous regions, but in non-mountainous regions, the richness is mostly affected by the precipitation seasonality, while there is no regularity in the relationship between endemism and environmental factors. In addition, the richness model in mountainous regions was in conformity with the results of GAM.

## 1. Introduction

Exploration of biodiversity patterns is of great significance for biogeographical research and biodiversity conservation [1,2,3]. It is also crucial to clarify the patterns of species richness and endemism and the driving forces behind these phenomena. Elucidation of the mechanisms underlying the macro patterns of species richness can help to explain species differentiation and conservation [4,5].

The Qinghai-Tibetan Plateau (QTP) is the largest and highest plateau in the world, including a number of large mountains such as the Himalayas, Hengduan Mountains, Kunlun Mountains, and Qilian Mountains. The strong uplift and formation of the QTP due to the collision between the Indian plate and the Eurasian plate is considered as the most important geological event in the Miocene-Pliocene period [6], which altered the natural environment of the plateau itself and had significant impacts on the surrounding environment and even the global climate [7,8,9,10]. As one of the hotspots for biodiversity research in the world, the QTP spans three biodiversity hotspots, including the mountains in south-west China, the Himalayas, and the Indian-Myanmar mountains [11]. The plateau is characterized by high richness of species (including many endemic species) due to its geological, climatic, and ecological diversity. Some studies of the diversity patterns in the QTP have demonstrated that most taxa are located in the margins of the plateau, such as birds, aphids, and vascular plants [11,12,13,14]. The richness patterns of birds in the QTP are mainly attributed to topographical heterogeneity and temperature amplitude [14]. So far, there have been relatively few studies of the diversity patterns of Diptera in the QTP, and more taxonomic groups are needed to clarify the effects of environmental factors on the biodiversity patterns in this region.

Comparable to an island, a mountainous region may be viewed as a relatively independent biogeographical unit. As an underlying driving force of ecological and evolutionary processes, climate can shape biodiversity patterns in both time and space within mountainous regions, which is different in non-mountainous regions and lowlands. The same environmental factor may have inconsistent influence on species richness and endemism in different regions due to complex topographic heterogeneity [15]. Currently, relatively little is known about the different effects of the same environmental factor on biodiversity between mountainous and non-mountainous regions.

Insects account for about 75% of animal species on the earth. Research on species diversity has been mostly focused on Coleoptera, Lepidoptera, Hymenoptera and Hemiptera [11,16,17,18,19]. The diversity pattern of Diptera remains poorly understood, even though it is the fourth largest order of Insecta. Tipulidae is one of the most diverse families of Diptera, and has 4463 known species worldwide [20], which are widely distributed in all geographical regions of the world. Tipulidae has higher endemism than vertebrates and butterflies in the Mediterranean region [21,22]. It is an important member of the aquatic and terrestrial biotic communities, and can be widely found in wetlands, grasslands, forests, and rice fields. The weak flight ability and low host influence make Tipulidae an ideal model for studying insect diversity. The adults barely feed and have weak flight abilities; the larvae are mainly saprophagous; and a few species are herbivorous and carnivorous. Tipulidae are mostly at lower trophic levels [23,24]. The species richness at different trophic levels has close association with the food web, and is directly driven by ecological factors at the adjacent trophic levels [25]. Compared with that of herbivorous insects, the diversity pattern of Tipulidae may be more significantly affected by soil factors and climate factors than by vegetation factors [26], and its distribution pattern and dispersal are directly related to geological events or climate changes. There have been many taxonomic studies of Tipulidae, but little research on the spatial pattern of Tipulidae [22,27,28]. More specifically, it remains unclear whether there is a general distribution pattern of Tipulidae in the QTP and adjacent areas, and whether the same environmental factor has different effects on the diversity patterns of Tipulidae in mountainous and non-mountainous regions.

In this study, we analyzed the species richness and endemic pattern of Tipulidae in the QTP and its surrounding areas, established a multiple regression model for the species richness of Tipulidae in mountainous regions, and discussed the effects of environmental factors on the diversity of Tipulidae in mountainous and non-mountainous regions.

## 2. Materials and Methods

### 2.1. Study Area and Distribution Data

The study area was the QTP and its surrounding areas, including China, Vietnam, Laos, Myanmar, Bangladesh, India, Bhutan, Nepal, Pakistan, Afghanistan, Kyrgyzstan, Uzbekistan, Tajikistan, and Kazakhstan (Figure 1a).

The distribution data of Tipulidae were collected from three sources: (1) literature and collection sites for each species of Tipulidae based on the Catalogue of the Crane Flies of the World (https://ccw.naturalis.nl/ (accessed on 25 December 2021)); (2) specimen collection records from the Entomological Museum of China Agricultural University; and (3) distribution data from the Global Biodiversity Information Facility (https://www.gbif.org/ (accessed on 6 June 2022)) crawled by using R 4.1.2 and RStudio (package: “dismo”) (some data from Mongolia, Sri Lanka, South Korea and Russia were also included) [29]. The dataset mainly included the latest checklist and distribution information of Tipulidae in the QTP and its surrounding areas. The taxonomic information was further validated using the Catalogue of the Crane Flies of the World (https://ccw.naturalis.nl/ (accessed on 25 December 2021)). The duplicated and erroneous data were excluded, and the close distribution records of the same species were removed to avoid sampling bias. For the distribution data providing longitudes and latitudes in the original sources, the georeferenced data were retained directly. For those data only providing the names of the distribution sites, the geographic coordinates were determined by using the Google map. All distribution data were kept at the county or below level. Finally, 2589 distribution datasets of 1219 Tipulidae species were obtained. The distribution data were divided into mountainous and non-mountainous groups according to the Mountain Regions of Earth provided by the Center for Macroecology, Evolution and Climate GLOBE Institute (https://macroecology.ku.dk/ (accessed on 11 February 2022)) (Figure 1b).

### 2.2. Mapping of Species Richness and Endemism

For evaluation of species richness pattern, 1° grid scale was chosen for the final analysis. The fishnet tool was employed to divide the study area into 1° × 1° (~110 × 110 km) grids with the furthest distribution points as the boundary based on ArcGIS 10.2 (ESRI, Inc., Redlands, CA, USA). The number of species in each grid was counted, and the endemism of each grid was determined by weighted analysis method [30]. Species richness pattern and endemic pattern were displayed in different colors.

### 2.3. Environmental Data

As the larvae of most Tipulidae species feed on humus and only a few species feed on living plants, they are important decomposers [23,24]. Soil organic matter content is an important driver of decomposer abundance [31]. Compared with the diversity of strictly herbivorous insects, that of Tipulidae may be less correlated with vegetation. Hence, we chose the global soil organic matter content (GSOC) instead of the vegetation factor as the contributing factor [26]. The data of GSOC were obtained from ISRIC-World (https://www.isric.org/ (accessed on 11 January 2022)). Larger-scale patterns of species diversity are generally considered to be closely related to abiotic factors [32]. In order to explore how the environmental factors drive the diversity pattern of Tipulidae in the QTP and its surrounding areas, we selected 19 climatic factors (Abbreviations: Bio1 = Annual Mean Temperature; Bio2 = Mean Diurnal Range (max temp—min temp); Bio3 = Isothermality (Bio2/Bio7) (×100); Bio4 = Temperature Seasonality (standard deviation ×100); Bio5 = Max Temperature of Warmest Month; Bio6 = Min Temperature of Coldest Month; Bio7 = Temperature Annual Range (Bio5-Bio6); Bio8 = Mean Temperature of Wettest Quarter; Bio9 = Mean Temperature of Driest Quarter; Bio10 = Mean Temperature of Warmest Quarter; Bio11 = Mean Temperature of Coldest Quarter; Bio12 = Annual Precipitation; Bio13 = Precipitation of Wettest Month; Bio14 = Precipitation of Driest Month; Bio15 = Precipitation Seasonality (Coefficient of Variation); Bio16 = Precipitation of Wettest Quarter; Bio17 = Precipitation of Driest Quarter; Bio18 = Precipitation of Warmest Quarter; Bio19 = Precipitation of Coldest Quarter) in the bioclimatic variables from the World Meteorological Database (http://www.worldclim.org/ (accessed on 13 November 2021)) [33], with the spatial resolution of 30 s. For environmental heterogeneity, we selected latitude, longitude, above mean sea level (AMSL) and topographic heterogeneity as the factors. For topographic heterogeneity, we selected elevation range (elev-range) and elevation standard deviation (elev-stdev) [14]. The elevation data were obtained from the World Meteorological Database. Environmental data of all grids with a spatial resolution of 1° × 1° were extracted by calculating the average of each grid cell. Data extraction was performed using the software of ArcGIS 10.2 (ESRI, Inc., Redlands, CA, USA).

### 2.4. Data Analysis

We used Mountain Regions of Earth (https://macroecology.ku.dk/ (accessed on 11 January 2022)) as the template and crop tool in ArcGIS to distinguish the 247 grids for the mountainous group and 111 grids for the non-mountainous group. The richness index and center coordinates of each grid were exported by ArcGIS, and the endemic index was calculated by Excel’s COUNTIF and SUMIF functions matched with the grid, where each distribution point was located to center coordinates by the ROUNDDOWN and VLOOKUP functions. To determine whether there are significant differences in the environmental factors driving the diversity pattern of Tipulidae in mountainous regions and non-mountainous regions, the data of both groups were analyzed using the paired normality test. If the data followed a normal distribution, the “MASS” package of “R” was used to conduct a variance homogeneity test and independent sample t-test to judge the difference [34]; otherwise, a Wilcoxon rank sum test was performed, and the significance level was set at *p* < 0.05.

In order to test the correlation of all environmental variables with the richness and endemism of Tipulidae in mountainous regions and non-mountainous regions, the “ggcorrplot” package of “R” was used to screen the factors highly correlated with richness and endemism [35]. The significant level of correlation was set as *p* < 0.05. 

The screened environmental factors related to species richness and endemism were fitted by the generalized additive model (GAM) using the “mgcv” (Wood) package in “R”, respectively. When there was an ambiguous relationship between the explanatory variables and response variables, the GAM was used to determine whether there was a non-linear relationship between the variables [36,37]. In this study, when the value of K = 4, and the goodness of fit reached *p* < 0.05, the model was taken as optimal, and an adjusted R^2^ was used to quantify the variance interpretation rate of the environmental variable.

Finally, the Pearson correlation coefficients of all factors were tested to establish a prediction model for the richness pattern of Tipulidae in the QTP and its surrounding areas. Due to the involvement of many independent variables, stepwise multiple regression analysis was used to explore the relationships between multiple variables, reveal the collinearity among factors with significant influence, and verify the quality of the model according to the coefficients of the screened factors in the fitted regression equation.

## 3. Results

### 3.1. Richness and Endemism Patterns of Tipulidae

For all of the 1219 Tipulidae species, more species were distributed in the eastern region of the QTP and southern China, while there were few species in the northern edge and the main surface of the plateau. Higher richness of Tipulidae was found in the following areas: the Yadong-Sikkim region, the Kamen River Basin, the southeastern Tibet region, the Assam region, western Sichuan, Wuyi mountain, Qilian Mountain, Tianmu Mountain and Taiwan Island (Figure 2a) (Index of richness > 24). Most of these areas were also of high endemism. In addition, Gaoligong Mountains, Nanling Mountain, Qilian Mountain, Tianmu Mountain, Shennongjia Mountain, Indian Western Ghats, Uttarakhand, Western Ust-Kamenogorsk of Kazakhstan, Chiang Mai of Thailand were also highly endemic (Figure 2b) (Index of endemism > 7).

### 3.2. Correlation of Environmental Factors with Tipulidae Richness and Endemism

Based on the results of our Wilcoxon rank sum test, Bio1, Bio4, Bio5, Bio7, Bio8, and Bio10 in mountainous regions had significantly lower values than those in non-mountainous regions (*p* < 0.01), while it was opposite for Bio3, Bio18, GSOC, elev-range, elev-stdev, and AMSL (*p* < 0.01). The remaining factors showed no significant difference (*p* > 0.01) (Appendix A).

The GAM analysis results revealed that the species richness patterns of Tipulidae are highly correlated with its endemic patterns in the QTP and its surrounding countries and area) (*p* < 0.001). However, they were affected by different factors, and each factor might have different relative importance for species richness and endemism. Rainfall factors (Bio 12–19) and topographic heterogeneity factors were the most important factors, while energy factors (Bio 1–11) were the least relevant factors (Figure 3).

In mountainous regions, the species richness pattern was significantly associated with Bio18, elev-range, and elev-stdev (*p* < 0.001), and related to Bio12, Bio13, and Bio16 (*p* < 0.01). The endemic pattern was significantly associated with Bio12, Bio13, Bio16, Bio18, elev-range, and elev-stdev (*p* < 0.001) and related to Bio7 (*p* < 0.01) (Figure 3a).

In non-mountainous regions, the species richness pattern was associated with Bio15 (*p* < 0.01) and no factor was significantly associated with the endemism (Figure 3b). Notably, some factors with close values in the two groups showed significant correlations in mountainous regions but extremely low correlations in non-mountainous regions with species richness and endemism.

### 3.3. Effects of Environmental Factors in Mountainous and Non-Mountainous Regions

According to the results of R-sq.(adj) and *p*-value (Appendix A), the species richness of Tipulidae in the QTP and its surrounding areas was significantly influenced by Bio2, Bio7, Bio12, Bio13, Bio14, Bio15, Bio16, Bio18, elev-range, and elev-stdev.

Bio12, Bio13 and Bio16 had strong explanatory power for the richness of Tipulidae in mountainous regions (R^2^ = 0.0301, *p* = 0.00365; R^2^ = 0.0344, *p* = 0.00955, R^2^ = 0.0339; *p* = 0.00663). The species richness of Tipulidae increased with the annual rainfall and rainfall in the rainy season. However, these two factors had no explanatory power on species richness in non-mountainous regions (R^2^ = −0.0055, *p* = 0.527; R^2^ = −0.0089, *p* = 0.861; R^2^ = −0.00917, *p* = 0.999) (Figure 4a,b,d). Bio15 had a less significant negative effect on Tipulidae species richness in non-mountainous regions (R^2^ = 0.0518, *p* = 0.00931), and had no explanatory power on species richness in mountainous regions (R^2^ = 0.00655, *p* = 0.292). With increasing variation coefficient of precipitation seasonality, the species richness would decrease in non-mountainous regions (Figure 4c). Bio18 had relatively higher explanatory power and influence on Tipulidae species richness in mountainous regions (R^2^ = 0.0581, *p* = 7.98 × 10^−5^). The richness of Tipulidae increased with the warmest quarter precipitation, while in non-mountainous regions, the GAM generated insignificant results (R^2^ = 0.00529, *p* = 0.482) (Figure 4e). Topographic heterogeneity affected species richness and endemism of Tipulidae. There were high correlations between elevation range and standard deviation (Figure 3a,b), and thus the results of GAM were almost consistent (Figure 4f,g). The species richness of Tipulidae in mountainous regions significantly increased with increasing topographic heterogeneity (R^2^ = 0.145, *p* = 13.99 < 2 × 10^−16^; R^2^ = 0.104, *p* = 5.03 × 10^−6^), but topographic heterogeneity had no explanatory power on species richness in non-mountainous regions (R^2^ = 0.0074, *p* = 0.464; R^2^ = 0.0166, *p* = 0.317). 

The endemic pattern of Tipulidae in the QTP and its surrounding areas was significantly influenced by Bio2, Bio4, Bio7, Bio12, Bio13, Bio14, Bio15, Bio16, Bio17, Bio18, elev-range, elev-stdev, and longitude (Appendix A).

In mountainous regions, Bio12, Bio13, Bio16, and Bio18 had relatively higher explanatory power and influence (R^2^ = 0.0447, *p* = 0.000495; R^2^ = 0.0467, *p* = 0.00199; R^2^ = 0.0459, *p* = 0.000591; R^2^ = 0.0632, *p* = 4.01 × 10^−5^), and Bio7 had relatively lower explanatory power for the endemism of Tipulidae (R2 = 0.0302, *p* = 0.00362) (Figure 5a). Overall, the endemism of Tipulidae in mountainous regions increased with the rainfall factors (Figure 5b–e). In non-mountainous regions, only Bio14, Bio15, and Bio17 had extremely weak effects on endemism, and no law could be found in the GAM results (R^2^ = 0.0429, *p* = 0.0165; R^2^ = 0.0286, *p* = 0.0419; R^2^ = 0.0294, *p* = 0.0398) (Appendix A). The response pattern of Tipulidae endemism to topographic heterogeneity was almost the same as that of species richness (Figure 5f,g).

### 3.4. Prediction Model of Species Richness

In this study, *p*-values greater than 0.7 were considered as strong autocorrelation (Appendix A) [38]. A total of 13 representative factors were screened out through the Pearson correlation coefficients, including Bio2, Bio3, Bio4, Bio12, Bio14, Bio15, Bio18, Bio19, GSOC, elev-range, AMSL, longitude and latitude. We further performed stepwise multiple regression analysis on these 13 factors, and finally screened four main contributing factors, including Bio18, elev-range, longitude and Bio19. Since the species richness of Tipulidae fluctuated widely in the QTP and its surrounding areas, a logarithmic transformation was carried out on species richness, and a multiple regression equation was obtained based on correlation coefficients: y = −2.52766 + 0.000835 Bio18 + 0.000332 elev-range + 0.023789 longitude + 0.0012 Bio19 (R^2^ = 0.2183, F = 16.824) (Figure 6; Appendix A). Due to the strong autocorrelation of species richness and endemism and the same GAM results, we did not predict the model of the endemism in mountainous regions. The GAM had low explanatory power for the species richness and endemism in non-mountainous regions (Appendix A). Hence, we did not predict the model in non-mountainous regions.

## 4. Discussion

### 4.1. Diversity Pattern

Higher species richness of Tipulidae was found in the eastern regions of the QTP. Such a distribution pattern has also been observed in Hemiptera and birds [3,39]. These results confirm the important role of the QTP as a glacial refuge in Asia for most biological taxa [39,40,41]. The species richness of Tipulidae was concentrated in seven areas, which are mostly located in broad continuous mountains or block mountains in China and India.

In central and eastern China, the highly endemic areas of Tipulidae include Wuyi mountain (Fujian), Tianmu mountain (Zhejiang), and Shennongjia mountain (Hubei), which are mostly noncontiguous areas. In western China, the highly endemic areas of Tipulidae form a large continuous area, from western Sichuan, Gaoligong Mountains to southeastern Tibet, and is even extended to northern India and Assam. Gongga Mountain, Kamen River, and Sikkim-Yadong were three areas with the highest endemism of Tipulidae in the QTP and its surrounding areas. In Gongga Mountain and Sikkim Yadong, Tipulidae can adapt to high elevation from 1500 m to 3500 m, and is distributed from low to high elevation (1000–2500 m) in the Kamen River Basin. We imagine that the high endemism of Tipulidae in the Gongga Mountains, Kamen River, and the Sikkim-Yadong areas could be attributed to a combination of environmental factors, historical events, and research depth. 

In the early 20th century, the British troops entered Tibet from the Sikkim-Yadong areas [42], and some western entomologists collected numerous specimens from Tibet [43]. In the 1930s, many missionaries (such as Armand David) and explorers found numerous biological resources in the Gongga Mountains [44]. The species of Tipulidae in the Kamen River Basin were systematically collected and published from 1961 to 1971 by Alexander [45,46]. Therefore, the high species richness and endemism in small-scale regions can be ascribed to a combination of different factors such as climate, topographic heterogeneity, and research history. The high diversity of Tipulidae suggests that dramatic speciation events might have occurred in these regions and indicates the complex research history and diverse ecological patterns around the QTP.

### 4.2. Correlation between Environmental Factors and Tipulidae Species Diversity

Environmental factors have inconsistent effects on the species richness of Tipulidae between mountainous and non-mountainous regions in the QTP and its surrounding areas (Figure 4 and Figure 5, Appendix A), suggesting that the richness pattern of Tipulidae is also affected by environmental factors such as elevation, monsoon, and microclimate [47,48]. Precipitation and topographic heterogeneity are the most important predictors of the diversity of Tipulidae. Although soil organic matter content is significantly different between mountainous and non-mountainous regions, it is largely irrelevant to the species richness and endemism of Tipulidae.

Most energy factors significantly differ between mountainous and non-mountainous regions (Appendix A). However, it has no significant effect on the diversity of Tipulidae. The diversity pattern of Tipulidae in QTP and its surrounding areas showed that low-latitude regions have higher species richness than high-latitude regions. Hydrodynamic theory has predicted that species richness is primarily determined by the availability of water and environmental energy: the former has a more significant effect on low-latitude regions and the latter has a greater effect on high-latitude regions of the northern hemisphere [32]. Besides, the diversity of Tipulidae is less affected by warmer climates, which confirms the previous finding that Tipulidae has strong freezing resistance and can use diverse habitats for adaptation to temperature changes [49].

Tipulidae is mostly terrestrial and prefers humid habitats with a weak tolerance to drought [49]. The rich precipitation in the warmest season results in a gradual increase in the richness of Tipulidae in mountainous regions, while in non-mountainous areas, precipitation in the warmest season has no explanatory power for the richness of Tipulidae. The rainfall in the warmest season of mountainous regions was significantly higher than that of non-mountainous regions (Appendix A), which may be an important driving factor for the higher richness of Tipulidae in mountainous regions.

Precipitation seasonality has a greater impact on the diversity of Tipulidae than other factors in non-mountainous regions, while it shows no obvious effect in mountainous regions. A high variation coefficient of precipitation seasonality indirectly represents instability and drastic changes of weather, which will cause a significant reduction of species diversity. However, the special environmental conditions in mountainous regions can somewhat alleviate the negative influence of dramatic weather changes, thereby better protecting the species diversity.

Climatic factors, rather than topographic factors, have the strongest explanatory power for species richness of birds, mammals, amphibians and bats [50], but topographic heterogeneity was also found to be crucial for the richness of birds in some studies [51]. Our results showed that the richness of Tipulidae sharply increases with elevation range over 4000 m, which may be associated with the special geographical conditions on the edge of the QTP; these regions tend to have very large elevation ranges and diverse climates.

Topographic heterogeneity only showed a significant effect on the species richness of Tipulidae in mountainous regions (Figure 4). In non-mountainous regions, there are also some mountains as indicated by the elevation range, but these mountains are discrete and isolated and therefore have little mountain-mass effect [52]; they can hardly play a full role as a “bridge” or “shield” [53,54,55] to weaken the impact of extreme climates to create stable microclimates and diverse habitats [50,56]. Generally, climatic stability is a key factor for high species richness in an area [57], and precipitation seasonality is more influential than topographic heterogeneity in non-mountainous regions. However, the mountain-mass effect can be enhanced by massive mountains in mountainous regions, and the warmest quarter precipitation and topographic heterogeneity are the most important factors.

In terms of endemism, topographic heterogeneity also has less significant impacts on Tipulidae in non-mountainous regions, while in mountainous regions, the greater topographic heterogeneity may bring about more complex and diverse resource heterogeneity and endemic climates [58], resulting in an increase in the endemism of Tipulidae.

In this study, the prediction model only has an interpretability of 21.83%, and therefore more factors should be considered in future studies, such as human factors and local microclimate.

## 5. Conclusions

In this study, the diversity of Tipulidae was firstly analyzed in the QTP and its surrounding areas. Three areas had high diversities of Tipulidae around the QTP, including the Sikkim-Yadong area, Kamen River Basin, and Gongga Mountain. It was found that the warmest quarter precipitation and topographic heterogeneity had significant effects on the diversity of Tipulidae in mountainous regions, and precipitation seasonality negatively affected its richness in non-mountainous regions. We speculate that Tipulidae may have sensitive responses to local microclimate, which may be a key factor driving the differentiation of Tipulidae in mountainous regions.

## Figures and Tables

**Figure 1 insects-13-01054-f001:**
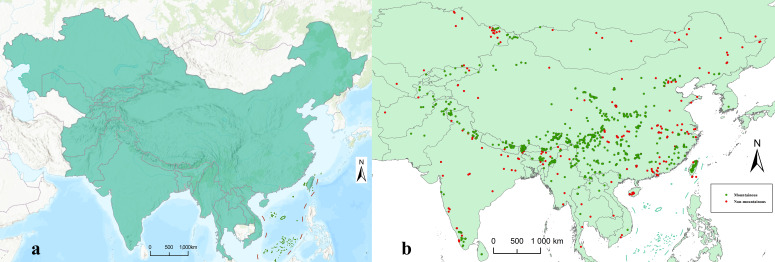
(**a**) Study area (malachite green part). (**b**) Distribution data of Tipulidae in the Qinghai-Tibet Plateau and its surrounding areas (Some data from Mongolia, Sri Lanka, South Korea and Russia were also included, but outside the main study areas). Green and red points represent distribution in mountainous and non-mountainous regions, respectively.

**Figure 2 insects-13-01054-f002:**
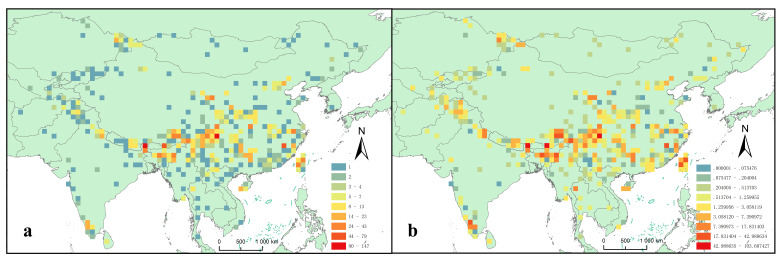
(**a**) Species richness pattern and (**b**) Endemic pattern of Tipulidae in the Qinghai-Tibet Plateau and its surrounding areas. (The numbers in (**a**) are the species richness indices, which represent the number of species; the numbers in (**b**) are the endemic index, which represent the sum of the inverses of the number of grids occupied by all species in the grid.) Dimensions of grids at 1° × 1° (~110 × 110 km).

**Figure 3 insects-13-01054-f003:**
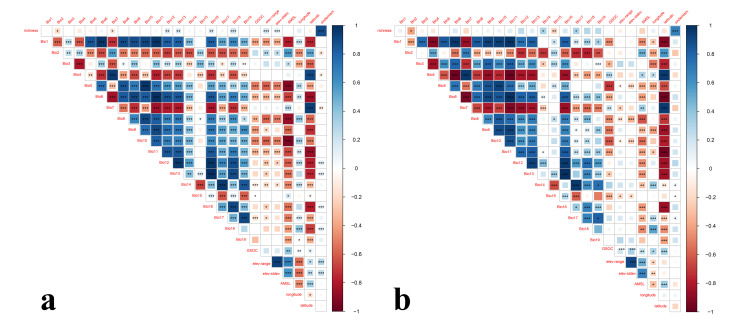
Correlation analysis of environmental factors with Tipulidae richness and endemism in the Qinghai-Tibet Plateau and its surrounding area. (**a**) Mountainous regions; (**b**) Non-mountainous regions. Warm colors represent negative correlation, cool colors represent positive correlation, and the darker the color, the higher the correlation. The significant level of correlation: * *p* < 0.05, ** *p* < 0.01, *** *p* < 0.005.

**Figure 4 insects-13-01054-f004:**
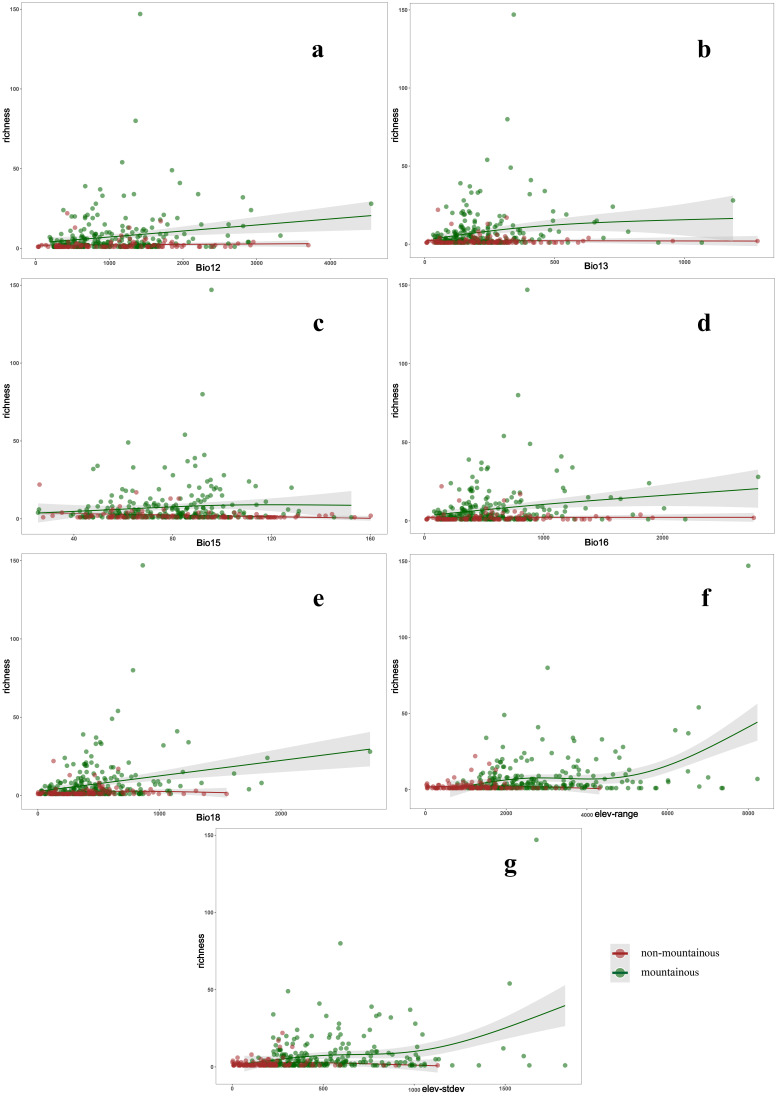
Relationships between richness and environmental factors based on the generalized additive model in the Qinghai-Tibet Plateau and its surrounding area. (**a**) Bio12: Annual Precipitation; (**b**) Bio13: Precipitation of Wettest Month; (**c**). Bio15: Precipitation Seasonality (Coefficient of Variation); (**d**) Bio16: Precipitation of Wettest Quarter; (**e**) Bio18: Precipitation of Warmest; (**f**) Elev-range: elevation range; and (**g**) Elev-stdev: elevation standard deviation. Gray borders around the lines indicate confidence intervals.

**Figure 5 insects-13-01054-f005:**
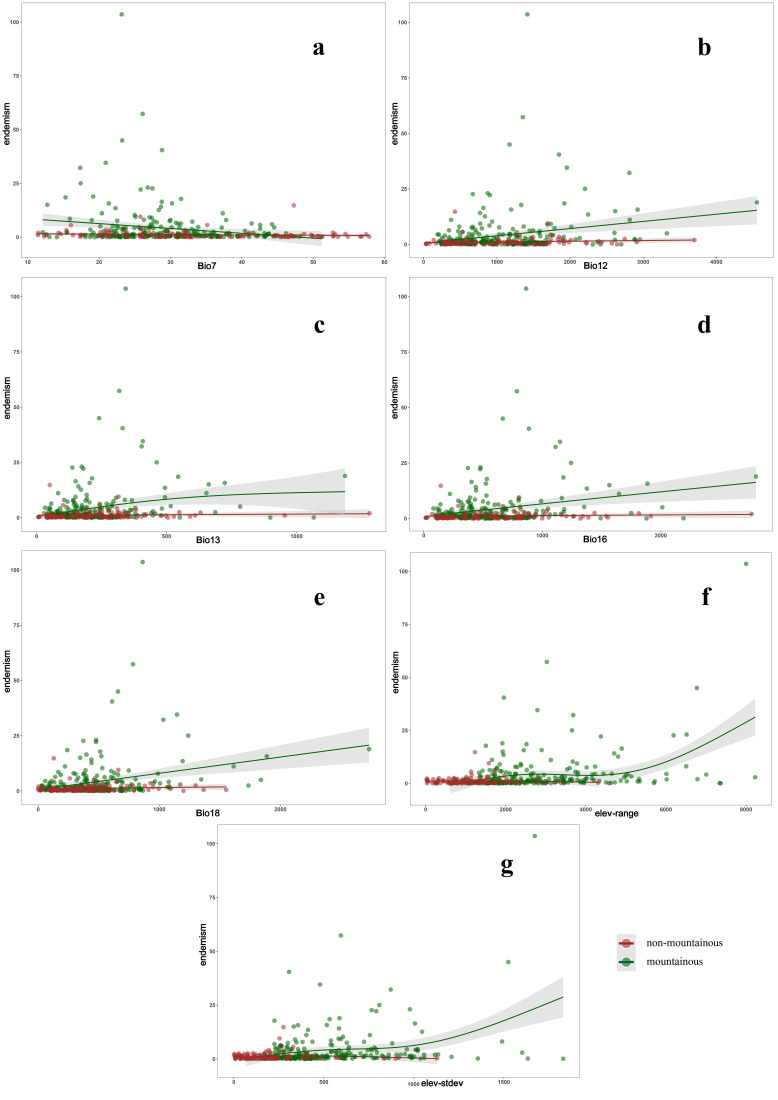
Relationships between endemism and environmental factors based on the generalized additive model in the Qinghai-Tibet Plateau and its surrounding area. (**a**) Bio7: Temperature Annual Range; (**b**) Bio12: Annual Precipitation; (**c**) Bio13: Precipitation of Wettest Month; (**d**). Bio16: Precipitation of Wettest Quarter; (**e**) Bio18: Precipitation of Warmest; (**f**) elev-range: elevation range; and (**g**) elev-stdev: elevation standard deviation. Gray borders around the lines indicate confidence intervals.

**Figure 6 insects-13-01054-f006:**
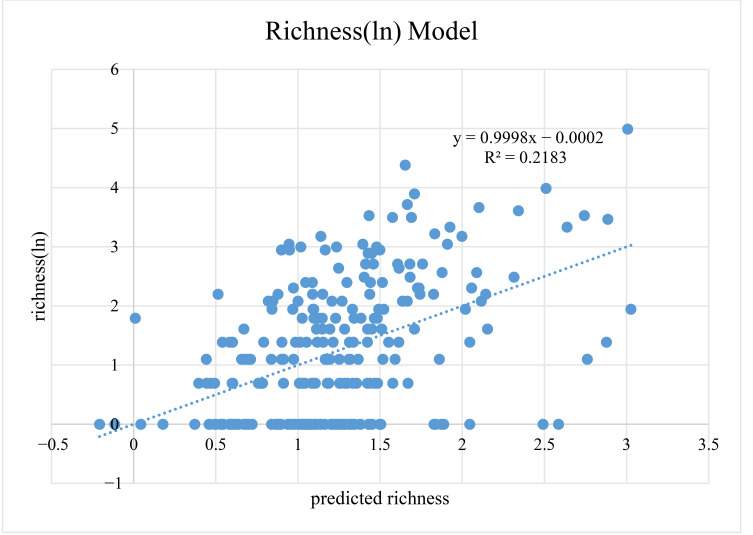
Richness (natural logarithm) model of Tipulidae in mountainous regions of the Qinghai-Tibet Plateau and its surrounding areas.

## Data Availability

The data presented in this study are available upon request from the corresponding author. The data are not publicly available due to the uncompleted subject.

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
