# Peer review of "The Effect of Environmental Factors on the Diversity of Crane Flies (Tipulidae) in Mountainous and Non-Mountainous Regions of the Qinghai-Tibet Plateau and Surrounding Areas"

_insects, 2022, doi:10.3390/insects13111054_

Round 1
Author Response
Dear Reviewer,
Thank you for the reviewers’ comments concerning our manuscript entitled “Diversity of crane flies (Tipulidae) in mountainous and non-mountainous regions of the Qinghai-Tibet Plateau and sur-rounding areas” (insects-1990364). Those comments are valuable and very helpful. We have read through comments carefully and have made corrections. Based on the instructions provided in editor’s letter, we uploaded the file of the revised manuscript. The responses to the reviewer's comments presented following.
Lines 45-48: Exploration of biodiversity patterns and species richness and endemism is of great significance for biogeographical research and biodiversity conservation [1-3]. It is also crucial to clarify the patterns of species richness and endemism and the laws behind these phenomena.
I would change for: Exploration of biodiversity patterns is of great significance for biogeographical research and biodiversity conservation [1-3]. It is also crucial to clarify the patterns of species richness and endemism and the laws behind these phenomena.
Response: modified.
Fig. 1b: Mongolia should lay also inside malachite green part(?)
Response: Some data from Mongolia, Sri Lanka, South Korea and Russia were also included, but outside the main study areas (line 108-110).
Fig. 3. the text for the image should be more explanatory (e.g. the meaning of the colours).
Response: Added.
It is easy to understand that elevation range in extremely high values implies high both degree of endemism and species richness. However, how to explain the same in several correlations between those characters and meteorological data (Bio12, 16, 18)?
Response: This is discussed in the third paragraph of Section 4.2 (line 375-381).
Line 345: Alexender should be Alexander
Response: Modified.
Lines 414-5: “It was found that the richness and endemism of Tipulidae are affected by a combination of environmental factors and historical events. “ I have not found any variable (chapter 2.3) referring to historical events (in spite they should have substantial effect). I mean that this sentence should be reformulated.
It seems to me that many times the differences between sites and results of analyses are generally due to different degrees of exploration of tipulid fauna of the areas rather than the actual influence of environmental factors (e.g. Mongolia, having relatively low richness but higher endemism; similar situation is that very low number of data originated from Central Tibetan plateau as apparent from Fig. 2). Research intensity is mentioned several times in the text as an important factor, but I have not found any attempt to interpret this factor (i.e. how it could affect, for example, outliers in correlation relationships)
Response: It is difficult for us to quantify the degree of research. Most studies of insect diversity may face this problem. We acknowledge the impact of research history on the results and removed relevant content from the conclusions section to the discussion section.
We would love to thank you for allowing us to resubmit the revised copy of the manuscript and we highly appreciate your time and consideration.
Best wishes
Yours sincerely
Qicheng Yang

Reviewer 2 Report
Authors have prepared an interesting analysis of tipulid diversity in a broad area of the Palearctic Region, focusing on the Qinghai-Tibetan Plateau, using literature and museum records. The study has merit, particularly if the following concerns are addressed:
Title: Add “crane flies (Tipulidae)” to the title and shorten “surrounding countries and areas” to “surrounding areas”. Please reduce “surrounding countries and areas” to “surrounding areas” throughout the manuscript.
The last sentence of the Simple Summary and Abstract are rather empty. Please briefly state how the results provide “potent evidence” of species diversity and how the results provide “useful reference” and “basic data” and the value of this information.
Better keywords are needed. Tipulidae and QTP are in the title and, therefore, not needed as keywords.
Line 47: There are no laws in biology. Use “driving forces”.
Throughout manuscript: Delete superlatives such as “detailed”, “great”, “potent”, etc. Readers, not authors, should determine if these terms apply to the study.
Global soil organic matter content as “the contributing factor” is a very coarse-grain measure of habitats, so it is not surprising that the results indicate it is irrelevant for richness and endemism. Can authors give more justification for selecting this factor rather than some other coarse-grain factor such as vegetation cover or land use.
Lines 208-210 and 317-321: Authors conclude that certain areas of the QTP have more species. However, how do the authors deal with sampling bias (i.e., the more records the more species)? The possibility of sampling bias is mentioned only briefly as “research depth” (lines 333 and 347). Can authors perform an analysis to determine if species richness is related to having more records for an area?
All figures are difficult to interpret because the figure captions are not adequately explained. Please use the figure captions to tell the reader how to interpret the figures. For example, in Fig. 2, the meaning of the numbers beside the different color boxes needs to be explained. Are the numbers in Fig. 2a species? If so, please state this in the caption. Also state what the numbers in Fig. 2b indicate. Figure 3 and all the other figures also need more explanation to enable interpretation.
Captions for Figures 4 and 5 are not correct. Figure 4 shows richness but the caption says endemism. Figure 5 shows endemism but the caption says richness. Please explain the gray borders around the lines in Figures 4 and 5.
Line 224: What is SPSS? Are authors referring to the statistical software SPSS? If so, the authors need to be more specific about what analyses (not software) the results are based on.
Line 232: Please explain what are “energy factors”? This is the first time this term is used in the manuscript.
Lines 329-330 and elsewhere: “elevation” not “altitude”
Lines 384 and 391: What are “special” environmental and geographical conditions? Please clearly explain.
Conclusions don’t say much. Please make the Conclusions more informative and stronger.
Author Response
Dear Reviewer,
Thank you for the reviewers’ comments concerning our manuscript entitled “Diversity of crane flies (Tipulidae) in mountainous and non-mountainous regions of the Qinghai-Tibet Plateau and sur-rounding areas” (insects-1990364). Those comments are valuable and very helpful. We have read through comments carefully and have made corrections. Based on the instructions provided in editor’s letter, we uploaded the file of the revised manuscript. The responses to the reviewer's comments presented following.
Title: Add “crane flies (Tipulidae)” to the title and shorten “surrounding countries and areas” to “surrounding areas”. Please reduce “surrounding countries and areas” to “surrounding areas” throughout the manuscript.
Response: Done.
The last sentence of the Simple Summary and Abstract are rather empty. Please briefly state how the results provide “potent evidence” of species diversity and how the results provide “useful reference” and “basic data” and the value of this information.
Response: Modified. (Line 20-24)
Better keywords are needed. Tipulidae and QTP are in the title and, therefore, not needed as keywords.
Response: Done.
Line 47: There are no laws in biology. Use “driving forces”.
Response: Done. (Line 45)
Throughout manuscript: Delete superlatives such as “detailed”, “great”, “potent”, etc. Readers, not authors, should determine if these terms apply to the study.
Response: Done.
Global soil organic matter content as “the contributing factor” is a very coarse-grain measure of habitats, so it is not surprising that the results indicate it is irrelevant for richness and endemism. Can authors give more justification for selecting this factor rather than some other coarse-grain factor such as vegetation cover or land use.
Response: Done and added the literature. (Line 145-146)
Lines 208-210 and 317-321: Authors conclude that certain areas of the QTP have more species. However, how do the authors deal with sampling bias (i.e., the more records the more species)? The possibility of sampling bias is mentioned only briefly as “research depth” (lines 333 and 347). Can authors perform an analysis to determine if species richness is related to having more records for an area?
Response: We acknowledge the impact of research history on the results, but it is difficult for us to quantify the degree of research. Most studies of insect diversity may face this problem. The members of Professor Yang Ding’s research group and the first author have carried out the investigation and research on crane flies for many years in the Qinghai-Tibet Plateau and surrounding areas. So we have a good data base of Tipulidae. We can only try to add survey data to reduce the sampling bias.
All figures are difficult to interpret because the figure captions are not adequately explained. Please use the figure captions to tell the reader how to interpret the figures. For example, in Fig. 2, the meaning of the numbers beside the different color boxes needs to be explained. Are the numbers in Fig. 2a species? If so, please state this in the caption. Also state what the numbers in Fig. 2b indicate. Figure 3 and all the other figures also need more explanation to enable interpretation.
Response: The numbers in fig. 2a are the richness indices, which represent the number of species. The numbers in fig. 2b are the endemic index, which represent the sum of the inverses of the number of grids occupied by all species in the grid. Its calculation method is in line 137, 176-179. Supplementary explanations have been made in figure caption.
Captions for Figures 4 and 5 are not correct. Figure 4 shows richness but the caption says endemism. Figure 5 shows endemism but the caption says richness. Please explain the gray borders around the lines in Figures 4 and 5.
Response: Modified. Gray borders around the lines indicate confidence intervals (Added).
Line 224: What is SPSS? Are authors referring to the statistical software SPSS? If so, the authors need to be more specific about what analyses (not software) the results are based on.
Response: Wilcoxon rank sum test. Modified.
Line 232: Please explain what are “energy factors”? This is the first time this term is used in the manuscript.
Response: Rainfall factors (Bio 12 – 19), energy factors (Bio 1 – 11). Added.
Lines 329-330 and elsewhere: “elevation” not “altitude”
Response: Modified.
Lines 384 and 391: What are “special” environmental and geographical conditions? Please clearly explain.
Response: Added. (Line 394-395).
Conclusions don’t say much. Please make the Conclusions more informative and stronger.
Response: Modified.
We would love to thank you for allowing us to resubmit the revised copy of the manuscript and we highly appreciate your time and consideration.
Best wishes
Yours sincerely
Qicheng Yang

Round 2
Reviewer 2 Report
The authors have improved the manuscript according to previous recommendations. However, a few cleanup items are still needed as follows:
Simple summary: Authors need to make three separate sentences rather than one long, grammatically incorrect sentence. The sentences should be corrected as follows: “Our results revealed three highly endemic regions and provided a species richness model of a group of decomposer insects. The warmest quarter precipitation and topographic heterogeneity have main effects on the diversity of Tipulidae in mountainous regions. These findings provide a reference for the diversity model of decomposers to aid protection of biodiversity in Asian mountains.”
Lines 155-157: Authors should correct the sentence as follows, or it does not make sense. The correct sentences should read as follows: “Soil organic matter content is an important driver of decomposer abundance [32]. Compared with the diversity of strictly herbivorous insects, that of Tipulidae may be less correlated with vegetation.”
Figure captions for Figures 3, 4, 5, and 6 require more information. Please add “in the Qinghai-Tibet Plateau and its surrounding area.”
Figure 7: The figure caption requires more information. Please explain what the figure is showing, such as what aspects are shown for the model of Tipulidae (species richness?) and the meaning of the different colors.
Author Response
Dear Reviewer,
Thank you for your comments concerning our manuscript entitled “Diversity of crane flies (Tipulidae) in mountainous and non-mountainous regions of the Qinghai-Tibet Plateau and sur-rounding areas” (insects-1990364). Those comments are valuable and very helpful, we sincerely appreciate your help. We have read through comments carefully and have made corrections. Based on the instructions provided in editor’s letter, we uploaded the file of the revised manuscript. The responses to the reviewer's comments presented following.
Simple summary: Authors need to make three separate sentences rather than one long, grammatically incorrect sentence. The sentences should be corrected as follows: “Our results revealed three highly endemic regions and provided a species richness model of a group of decomposer insects. The warmest quarter precipitation and topographic heterogeneity have main effects on the diversity of Tipulidae in mountainous regions. These findings provide a reference for the diversity model of decomposers to aid protection of biodiversity in Asian mountains.”
Response: Done.
Lines 155-157: Authors should correct the sentence as follows, or it does not make sense. The correct sentences should read as follows: “Soil organic matter content is an important driver of decomposer abundance [32]. Compared with the diversity of strictly herbivorous insects, that of Tipulidae may be less correlated with vegetation.”
Response: Done.
Figure captions for Figures 3, 4, 5, and 6 require more information. Please add “in the Qinghai-Tibet Plateau and its surrounding area.”
Response: Done.
Figure 7: The figure caption requires more information. Please explain what the figure is showing, such as what aspects are shown for the model of Tipulidae (species richness?) and the meaning of the different colors.
Response: Done. Figure 7: MaxEnt forecasting model of Tipulidae based on 19 climate factors (AUC = 0.867). Contemporary rich-ness of Tipulidae in the Qinghai-Tibet Plateau and its surrounding area, the warmer the color, the higher the richness, shown that areas of high richness may have similar climates.
We would love to thank you for allowing us to resubmit the revised copy of the manuscript and we highly appreciate your time and consideration.
Best wishes
Yours sincerely
Qicheng Yang